# Preserving Health Beyond Infection Control: Frailty, Weight, and Cognition in OPAT Patients

**DOI:** 10.3390/antibiotics14111173

**Published:** 2025-11-20

**Authors:** Giacomo Ciusa, Giuseppe Pipitone, Bianca Catania, Giulia Coniglione, Claudia Imburgia, Maria Grazia Laura Marsala, Preziosa Scordo, Antonio Albanese, Antonio Cascio, Giovanni Guaraldi, Chiara Iaria

**Affiliations:** 1Infectious and Tropical Diseases Unit, A.R.N.A.S. Civico-Di Cristina-Benfratelli, 90100 Palermo, Italy; 2Department of Health Promotion, Mother and Child Care, Internal Medicine and Medical Specialties “G D’Alessandro”, University of Palermo, 90100 Palermo, Italy; antonio.cascio03@unipa.it; 3Infectious and Tropical Disease Unit, AOU Policlinico “P. Giaccone”, 90100 Palermo, Italy; 4Infectious Diseases Unit, Hospital Papardo, 98120 Messina, Italy; giulia.coniglione94@gmail.com (G.C.); preziosascordo@aopapardo.it (P.S.);; 5Quality and Clinical Risk Management Unit, A.R.N.A.S. Civico-Di Cristina—Benfratelli, 90100 Palermo, Italy; 6Modena HIV Metabolic Clinic, Department of Surgical, Medical, Dental and Morphological Sciences, University of Modena and Reggio Emilia, 41100 Modena, Italy

**Keywords:** Outpatient Parenteral Antimicrobial Therapy (OPAT), multidrug-resistant (MDR) bacteria, frailty, body weight, subjective cognitive status

## Abstract

**Background:** Outpatient Parenteral Antimicrobial Therapy (OPAT) is a validated alternative to inpatient care for complicated infections, ensuring clinical efficacy, safety, and cost-effectiveness. However, its impact on patient-centered outcomes such as nutritional status, frailty, and cognitive well-being has rarely been studied. **Methods:** We conducted a multicentric retrospective observational study of patients treated with OPAT between April 2024 and July 2025 in two tertiary care hospitals. Baseline demographics, comorbidities, weight, frailty status (Rockwood Clinical Frailty Scale (CFI)), and infection-related variables were collected. Follow-up assessments evaluated body weight, frailty, and subjective cognitive status. Clinical outcomes, adverse events, and hospital readmissions were recorded. **Results:** Of 139 patients treated with OPAT, 119 were included in the analysis (56% male, median age 67 years). Common comorbidities were ischemic heart disease (33%), diabetes mellitus (29%), chronic pulmonary disease (22%), and solid tumors (19%). The most frequent infections were urinary tract infections (UTIs) (29%), osteomyelitis (25%), and pneumonia (17%). Multidrug-resistant (MDR) organisms were isolated in 66% of cases. Clinical recovery occurred in 82,5% of patients, while 16% required readmission in the next 30 days; no deaths were reported. Body weight (median 73 vs. 73.0 kg at baseline, *p* = 0.43) and frailty index (median 2.5 vs. 2.4, *p* = 0.16) remained stable. Cognitive status was unchanged in 85.6%, declined in 5.9%, and improved in 8.5%. **Conclusions:** OPAT was confirmed to be clinically robust and well tolerated, with additional potential benefits in preserving weight, frailty status, and cognitive well-being. These findings suggest that OPAT not only ensures infection control but may also protect against hospitalization-related functional decline. Prospective studies incorporating standardized geriatric and cognitive assessments are needed to confirm these preliminary findings and define OPAT’s broader role in holistic patient care.

## 1. Introduction

Outpatient Parenteral Antimicrobial Therapy (OPAT) has emerged as a safe and effective alternative to prolonged hospital admission for the management of complicated or deep-seated infections. Large observational studies and meta-analyses have confirmed its clinical efficacy and safety [1,2,3], while health economic analyses demonstrated its cost-effectiveness and the ability to optimize healthcare resource utilization [4]. Patient satisfaction is generally high, reflecting the benefits of treatment at home and the avoidance of prolonged hospitalization [5]. Furthermore, the introduction of elastomeric pumps has expanded the spectrum of antibiotics suitable for OPAT, including time-dependent beta-lactams that are stable over 24 h [6], thereby broadening the eligibility criteria and improving prescription appropriateness.

However, despite extensive evidence on safety, efficacy, and economics, OPAT research has predominantly focused on infection control and reduction in hospital bed occupancy. There is a lack of systematic investigation into outcomes that reflect the patient’s overall health and quality of life. Hospitalization is known to increase risks of malnutrition, sarcopenia, frailty, and delirium, particularly in older adults or multimorbid patients [7,8,9,10,11,12,13]. Such multidimensional decline has become a central concern in the care of older adults with infection. These conditions strongly influence long-term survival and well-being, yet they are rarely included as endpoints in OPAT studies.

This gap underscores the need for a more comprehensive, patient-centered approach in OPAT evaluation. In particular, outcomes such as body weight, frailty status, and subjective cognitive performance may provide valuable insight into how OPAT impacts global health, beyond the traditional measures of infection resolution and hospital days saved. This study therefore aimed to expand the current understanding of OPAT by assessing not only infection-related outcomes but also indicators of global health and functional well-being.

## 2. Results

A total of 139 patients were treated in OPAT during the study period. A total of 20 were excluded due to incomplete data or loss to follow-up, leaving 119 patients for analysis.

Enrolled population were 67/119 (56.3%) male and 52/119 (43.7%) female, with a median age of 67 years (IQR 51–76). The median CCI Index was 3 (IQR 2–6). The study population reflected a multimorbid and predominantly older cohort, consistent with the typical clinical profile of patients eligible for OPAT. The most frequent comorbidities were ischemic heart disease (*n* = 39/119, 32.8%), diabetes mellitus (*n* = 35/119, 29.4%), chronic obstructive pulmonary disease (*n* = 26/119, 21.8%), and solid tumors (*n* = 23/119, 19.3%). Median baseline body weight was 73 kg (IQR 67–79), and the median frailty index was 2 (IQR 2–3).

Infectious diagnoses included urinary tract infections (UTIs) (*n* = 35/119, 29.4%), osteomyelitis (*n* = 29/119, 25.2%), pneumonia (*n* = 20/119, 16.8%), and ABSSSI (*n* = 10/119, 8.4%). Overall, 92/119 (77%) patients showed microbiological isolate: the most common were *P. aeruginosa* (*n* = 27/92, 29.3%), *K. pneumoniae* (*n* = 23/92, 25%), and *S. aureus* (*n* = 15/92, 16.3%), with 66% classified as multi-drug resistant (MDR), including extended-spectrum beta-lactamase (ESBL), *Klebsiella pneumoniae* Carbapenemase (KPC), OXA-48 producing gram negatives, difficult-to-treat (DTR) *P. aeruginosa*, and methicillin-resistant *Staphylococcus aureus* (MRSA).

Baseline characteristics are listed in Table 1.

Treatment: A total of 88 patients received once-daily antibiotic regimens: the most frequent pathogens involved were *K. pneumoniae*, *S. aureus*, and *E. coli*; the most frequent agents administered were ceftriaxone, ertapenem, and daptomycin. A total of 31 patients were treated with continuous infusion via elastomeric pumps: the most frequent pathogens involved were *P. aeruginosa*, *K. pneumoniae*, *P. mirabilis*; the most frequent agents administered were ceftolozane/tazobactam and piperacillin/tazobactam. Adverse drug reactions occurred in 2.5% of cases, while catheter-related complications were observed in 12%. Treatment strategies, adverse drug reactions, and catheter-related complications are listed in Table 2.

Clinical outcomes: At the end of therapy, 98/109 (89.9%) patients had recovered, and 11/109 (10%) had a relapse, while 19/109 (16%) required hospital readmission, and no patient died.

At a follow-up visit performed within 30 days after the end of therapy, body weight, frailty, and subjective cognitive status were reassessed:Median body weight at follow-up was 73 kg (IQR 66–78), with no statistically significant variation compared to baseline (*p*-value 0.43). Body weight at follow-up was missing for 8 patients.Median frailty index at follow-up was 2 (IQR 2–3), with no significant variation compared to baseline (*p*-value 0.16). Frailty index was missing at follow-up for 20 patients.Subjective cognitive status: A total of 85.6% reported no change, 5.9% reported decline, and 8.5% reported improvement. Subjective cognitive status was missing at follow-up for 2 patients.

Clinical outcomes are shown in Figure 1 and Table 3.

## 3. Discussion

Efficacy and safety of OPAT have been extensively described in large observational studies [2,3]. Our findings further support its effectiveness across a wide range of infections, including those sustained by multi-drug-resistant (MDR) bacteria. Recent systematic reviews [1] reinforce that OPAT is not only safe but also clinically comparable to inpatient care, while being associated with high patient satisfaction and significant cost savings [2,4,5]. Beyond the standard once-daily antibiotic regimens, the use of elastomeric pump devices has expanded the therapeutic arsenal of OPAT, enabling the administration of 24 h stable beta-lactams [6]. This innovation broadens the spectrum of eligible patients and supports more appropriate antimicrobial prescriptions. In our cohort, patients treated with elastomeric pumps were frequently those with challenging infections, often involving MDR organisms, such as difficult-to-treat *Pseudomonas aeruginosa*. Their outcomes appeared comparable to those managed with once-daily regimens, with similar complication rates and clinical success, suggesting that elastomeric infusion is a reliable and safe strategy to optimize OPAT.

The burden of comorbidities in our population, as measured by the CCI, reflects the complexity of patients referred to OPAT. As patients selected to OPAT were those able to reach daily our ambulatories to receive antibiotic treatment, we observed low median frailty index and a cognitive functional status that allowed daily activities, despite high median age and elevated burden of comorbidities.

Despite this, adverse drug reactions and catheter-related complications remained limited, confirming the feasibility of OPAT even in frail and multimorbid individuals. Importantly, the implementation of OPAT in our hospitals resulted in the saving of 1635 hospital admission days over 15 months, generating significant economic benefits.

From a broader perspective, OPAT should be viewed as part of antimicrobial stewardship programs, which aim to reduce colonization by MDR organisms [14] and hospital-acquired infections [15]. Moreover, prescriptions are supervised by infectious disease specialists, ensuring appropriateness of antibiotic selection and duration.

To our knowledge, this is among the first studies to explore OPAT’s impact on frailty and cognitive health—dimensions that are increasingly recognized as key determinants of post-hospital outcomes.

In our cohort, weight and frailty index remained stable, suggesting OPAT may be associated with preservation of baseline health during treatment rather than contributing to the decline, as hospitalization does [11,12,13]. Thus, maintaining stability while treating an infection represents a goal to preserve long-term health and health-related quality of life.

Acute illness contributes to malnutrition and frailty [7], and hospitalization itself is a recognized risk factor for weight loss, sarcopenia, and functional decline [8,9], especially in older adults with longer stays and higher disability burden [10]. Weight loss and worsening frailty are strongly associated with increased mortality [11,12]. In our cohort, body weight and frailty index showed non-significant variation, suggesting that OPAT may help maintain patients’ baseline status during antimicrobial treatment. Similarly, hospitalization is a major trigger for delirium and long-term cognitive decline [13], whereas our patients, despite being treated for complicated infections with high-dose antibiotics, reported overall satisfactory cognitive outcomes. This emphasizes the importance of non-pharmacological aspects of OPAT, such as maintaining patients in their familiar home environment and reducing hospital-related stressors.

Protecting patients from malnutrition, frailty, and delirium typically requires the involvement of multidisciplinary teams, including nutritionists, physiotherapists, psychogeriatricians, and caregivers. OPAT, by contrast, streamlines care and appears to provide a simpler yet effective pathway to safeguard overall health while ensuring the appropriate use of antimicrobial therapy.

This study has several limitations. The specific organization of our OPAT service implies a selection of patients that were able to reach our ambulatories. This selection bias could limit the generalizability of our observations. Its retrospective nature and the absence of a control group restrict the strength of causal inference. The collection of frailty and cognitive outcomes was based on clinical evaluation and patient self-report, which introduces potential reporting bias. Furthermore, the relatively small sample size may limit the generalizability of the findings.

Our study also presents several strengths. First, it addresses an underexplored dimension of OPAT by evaluating not only infectious outcomes but also patient-centered parameters such as body weight, frailty status, and cognitive well-being. Second, it leverages real-world data from a large tertiary care hospital, reflecting daily clinical practice and including patients with complex comorbidities and MDR infections. Third, the use of both once-daily regimens and elastomeric pumps allowed us to assess a broad therapeutic spectrum, highlighting the feasibility of different OPAT modalities. Finally, by documenting the reduction in hospital admission days, our study contributes to the growing body of evidence on the role of OPAT in supporting health system sustainability.

This study was designed to investigate the OPAT potentialities beyond infection cure and to promote a broader understanding of health. Our aim was to assess well-being beyond hospitalization, considering both physical performance and psychological condition. The importance of new studies in this field lies in adopting a more holistic vision of the patient, in which the benefits of OPAT are not evaluated solely in terms of hospital days saved or economic costs, but in terms of health-related quality of life.

By integrating infection management with functional preservation, OPAT may represent a paradigm shift towards truly patient-centered infectious disease care.

## 4. Materials and Methods

This multicentric retrospective observational study included patients treated with OPAT between 1 April 2024 and 31 July 2025, either after hospitalization or as outpatients, in the Infectious Diseases Unit ambulatories of two Sicilian tertiary care hospitals (A.R.N.A.S. Civico-Di Cristina-Benfratelli of Palermo and A.O. Papardo of Messina). Patients were admitted to OPAT when affected by infection needing intravenous anti-biotic treatment, clinically stable and able to reach our ambulatory service. 

Antibiotic administration was conducted in accordance with the hospital’s protocols [16]. Peripheral venous catheters were used for once-daily antibiotic infusions, while elastomeric pumps with midline catheters allowed for the continuous infusion of time-dependent antibiotics that remained stable for 24 h [6].

Baseline data included demographic information (sex and age), comorbidities, body weight, body temperature, microbiological isolates, Charlson Comorbidity Index (CCI), Rockwood CFI, and laboratory results (white blood cell count and C-reactive protein). Frailty was assessed using the Rockwood CFI, ranging from 1 (very fit) to 9 (terminally ill); scores ≥ 5 indicated at least mild-to-moderate frailty.

Subjective cognitive status was evaluated at follow-up through a non-structured interview performed by trained clinicians, assessing self-perceived changes in memory, attention, and overall cognitive function.

Data on antibiotic prescriptions, duration of therapy, and adverse events were recorded. Clinical outcomes, assessed by trained infectious disease clinicians, were categorized as recovery, hospital readmission for any cause, or death. Adverse drug reactions and catheter-related complications were documented throughout follow-up.

Body weight was measured at baseline and follow-up using calibrated hospital scales. Frailty was reassessed at follow-up using the Rockwood CFI, as administered by trained clinicians. Subjective cognitive status was evaluated at follow-up through an interview, in which participants were asked whether they perceived no change, a decline, or improvement in memory, attention, or overall cognitive function and well-being compared to baseline. Responses were documented in the clinical chart and extracted for analysis.

Dichotomous variables are reported as absolute numbers and percentages. Continuous variables are presented as medians and interquartile ranges (IQR, 25–75%), with 95% confidence intervals.

A t-test was used for paired samples in the case of weight and Rockwood scale pre-to-post treatment variation analysis (both measures were normally distributed). A *p*-value ≤ 0.05 was considered statistically significant. For statistical analysis SPSS © (v. 21) was used.

## 5. Conclusions

This retrospective cohort study supports OPAT as an effective and safe alternative to hospital admission, with additional potential benefits in preserving body weight, frailty status, and cognitive well-being. These findings highlight OPAT not only as a vehicle for antimicrobial delivery but also as a model of care that may contribute to maintaining overall health and functional stability, particularly in older or multimorbid populations. Although the observed changes were modest, the study provides a foundation for refining outcome measures that better capture the multidimensional impact of OPAT on patient-centered health. Future prospective studies integrating standardized geriatric assessments, nutritional evaluation, and validated cognitive tools are warranted to confirm and expand these observations.

## Figures and Tables

**Figure 1 antibiotics-14-01173-f001:**
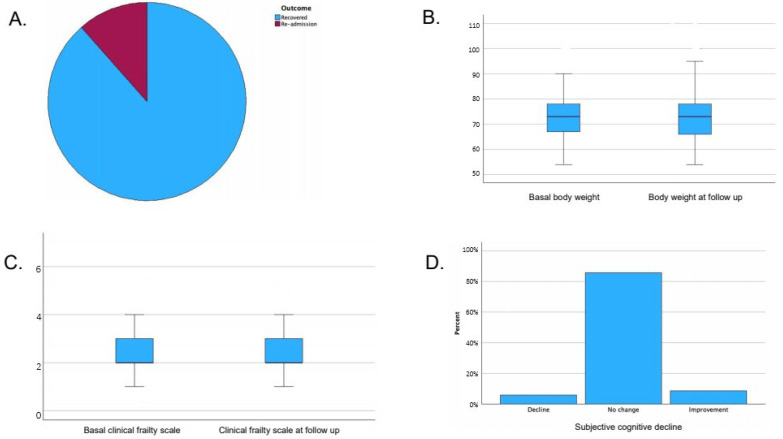
Clinical outcomes and follow-up assessments in the OPAT cohort. Panel (**A**): Distribution of clinical outcomes at the end of therapy (recovered, readmission). Panel (**B**): Box plots showing body weight variation from baseline to follow-up among recovered patients. Panel (**C**): Box plots showing Clinical Frailty Scale variation from baseline to follow-up among recovered patients. Panel (**D**): Bar chart showing patient-reported changes in cognitive status (no change, decline, improvement).

**Table 1 antibiotics-14-01173-t001:** Baseline characteristics, infectious diagnoses, microbiological isolates, and treatment regimens in the OPAT cohort. IQR: Inter Quartile Range; CODP: Chronic Obstructive Pulmonary Disease; UTI: Urinary Tract Infection; ABSSSI: Acute Bacterial Skin and Soft Tissues Infection; MDR: Multi-Drug Resistant; MRSA: Methicillin-Resistant *S. aureus*; ESBL: Extended-Spectrum Beta-Lactamase; KPC: *Klebsiella pneumoniae* Carbapenemase; DTR: Difficult-to-Treat.

Characteristic	Value
Number of patients	119
Male sex, *n* (%)	67 (56.3%)
Female sex, *n* (%)	52 (43.7%)
Age, median (IQR)	67 (51–76) years
Charlson Comorbidity Index, median (IQR)	3 (2–6)
Most frequent comorbidities	Ischemic heart disease (*n* = 39, 32.8%); Diabetes Mellitus (*n* = 35, 29.4%); COPD (*n* = 26, 21.8%); solid tumor (*n* = 23, 19.3%),
Baseline body weight, median (IQR)	73 kg (67–79)
Frailty Index, median (IQR)	2 (2–3)
Most frequent infectious diagnoses	UTIs (*n* = 35, 29.4%); Ostemyelitis (*n* = 29, 25.2%); Pneumonia (*n* = 20, 16.8%), ABSSSI (*n* = 10, 8.4%)
Most common microbiological isolates	*P. aeruginosa* (*n* = 27, 29.3%); *K. pneumoniae* (*n* = 23, 25%); *S. aureus* (*n* = 15, 16.3%),
MDR isolates, *n* (%)	61 (66%) including MRSA, ESBL, KPC, OXA-48, or DTR *P. aeruginosa*

**Table 2 antibiotics-14-01173-t002:** Treatment strategies, adverse drug reactions, and catheter-related complications.

Treatment	Value
Patients treated with once-daily regimens	88 (most frequent agents: *K. pneumoniae*, *S. aureus*, *E. coli*)
Patients treated with continuous infusion via elastomeric pumps	31 (most frequent agents: *P. aeruginosa*, *K. Pneumoniae*)
Adverse drug reactions, *n* (%)	3 (2.5%)-1 nausea, 1 *C. difficile* colitis
Catheter-related complications, *n* (%)	12 (10%)

**Table 3 antibiotics-14-01173-t003:** Clinical outcomes and follow-up parameters in the OPAT cohort. IQR: Inter Quartile Range.

Outcome	Value
Recovered, *n* (%)	98 (82.5%)
Hospital readmission, *n* (%)	19 (16%)
Deaths, *n* (%)	0 (0%)
Body weight at follow-up, median (IQR)	73 kg (66–78)
Change in body weight vs. baseline	Non-significant variation (*p*-value 0.43)
Frailty Index at follow-up, median (IQR)	2.5 (2–3)
Change in frailty vs. baseline	Non-significant variation (*p*-value 0.16)
Cognitive status—no change, *n* (%)	101 (85.6%)
Cognitive status—decline, *n* (%)	7 (5.9%)
Cognitive status—improvement, *n* (%)	10 (8.5%)

## Data Availability

Data is contained within the article.

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
