# Peer review of "Preserving Health Beyond Infection Control: Frailty, Weight, and Cognition in OPAT Patients"

_antibiotics, 2025, doi:10.3390/antibiotics14111173_

Round 1
Reviewer 1 Report
Comments and Suggestions for Authors
Overall, this manuscript requires further elaboration in several sections to improve clarity and understanding, particularly in the Introduction and Discussion.
Materials and Methods (Lines 200–203):
Please provide more details regarding the design of this retrospective observational study. Specifically, clarify the rationale for selecting only two tertiary hospitals and 119 patients for analysis. Is this sample size sufficient based on your statistical power or analysis plan?
Tables and Figures:
-
Figure 1 and Table 3: Additional explanation should be included in the Results section to help readers interpret these data more effectively.
Discussion:
Since the median age of the study population is 67 years, this demographic aspect should be discussed in relation to frailty, cognition, and overall health outcomes in OPAT patients.
Author Response
Comment 1: "Overall, this manuscript requires further elaboration in several sections to improve clarity and understanding, particularly in the Introduction and Discussion."
Response: Thank's to the reviewer for the observations that will help us to improve our manuscript. We extended Discussion section in the paragraphs about patient characteristic and study limitation, to explicitate the clinical setting of the study.
We thought the Introduction section was already sufficient to contestualize our observations.
Comment 2: "Please provide more details regarding the design of this retrospective observational study. Specifically, clarify the rationale for selecting only two tertiary hospitals and 119 patients for analysis. Is this sample size sufficient based on your statistical power or analysis plan?"
Response: We collected data during our ambulatory activity in the study period and we represeted them in a reptrspective cohort study. We included all patients afferent to our OPAT service, and sample size was not calculated.
Comment 3: "Additional explanation should be included in the Results section to help readers interpret these data more effectively."
Response: We agree with the observation and we included the clinical outcomes described in table 3 and figure in the full text in the section Results.
Comment 4: "Since the median age of the study population is 67 years, this demographic aspect should be discussed in relation to frailty, cognition, and overall health outcomes in OPAT patients."
Response: we agree with the observation. Discussion about basal demographic characteristic, basal frailty index and cognitive status were provided in Discussion section.
Thank's to the reviewer for the observation, we hope our anwers were satisfactory.
Best regards,
Giacomo Ciusa and Giuseppe Pipitone
Reviewer 2 Report
Comments and Suggestions for Authors
This manuscript addresses an original and clinically relevant topic, with specific attention to frailty, body weight, and subjective cognition. The paper is clearly written, well organized, and supported by appropriate references. The introduction provides a convincing rationale, highlighting the need for data on functional and cognitive aspects of OPAT, which are often neglected in previous research. The discussion is coherent and balanced, acknowledging limitations while emphasising the clinical implications. However, several methodological and reporting aspects could be improved.
The retrospective design and lack of a control group are appropriately stated as limitations, but their implications should be more explicitly discussed in relation to potential biases (selection bias, reporting bias). It would be useful to specify how patients were selected for OPAT and whether any criteria (clinical stability, social support, cognitive function at baseline) influenced eligibility, as these may affect generalizability. Clarification is needed on whether a standardized questionnaire was used, who performed the assessment, and whether the interview was validated or piloted.
The authors report p-values for weight and frailty comparisons but do not provide confidence intervals or effect sizes.
The discussion sometimes implies that OPAT preserves frailty and cognition; this wording may overstate causality. Given the study design, results can only support an association or lack of observed decline.
The manuscript mentions loss to follow-up but does not quantify missing data for frailty, weight, or cognition.
Despite these issues, the paper has merit and presents a novel perspective linking OPAT with functional preservation and quality of life. With a focused revision, it could make a meaningful contribution to the literature on holistic OPAT outcomes.
Author Response
Comment 1: "The retrospective design and lack of a control group are appropriately stated as limitations, but their implications should be more explicitly discussed in relation to potential biases (selection bias, reporting bias). It would be useful to specify how patients were selected for OPAT and whether any criteria (clinical stability, social support, cognitive function at baseline) influenced eligibility, as these may affect generalizability."
Response: thank's to reviewer comments and observations, that may help us to clarify our study. We agree with the first issue and we provided to explicitate the points in Material and Methods and Discussion sections.
Comment 2: "Clarification is needed on whether a standardized questionnaire was used, who performed the assessment, and whether the interview was validated or piloted."
Response: as the subjective cognitive status was assessed without standardized questionnaires, we modified the text according to the issue proposed.
Comment 3: "The authors report p-values for weight and frailty comparisons but do not provide confidence intervals or effect sizes."
Response: Confidence intervals and interquartile range are explicitated in Materials and Methods section.
Comment 4: "The discussion sometimes implies that OPAT preserves frailty and cognition; this wording may overstate causality. Given the study design, results can only support an association or lack of observed decline."
Response: We agree with the observation, and we modified the text according to the comment in Discussion section.
Comment 5: "The manuscript mentions loss to follow-up but does not quantify missing data for frailty, weight, or cognition"
Response: We agree with the comment and we explicitated the missin data in Results section.
Thank's to the reviewer for the useful observations, we hope that our answers andressed the issues.
Round 2
Reviewer 2 Report
Comments and Suggestions for Authors
Thank you for the revisions and the clarifications you have provided. I would also specify the pathogens involved in more detail.
Author Response
Comment 1: "specify the pathogens involved in more detail"
Response: Dear reviewer, the point you asked to clarify includes a large amount of results data and needs some consideration in discussion because the range of infectious diagnosis is wide, as soon as the microbiological speciments considered and the microbes isolated. Moreover, the extremely high number of multi-drugs resistent organisms found, that are consistent with the extremely high prevalence of multi-drug resistent organisms present in our country, deserves some specific consideration about clinical outcomes, choice of antibiotic and organizative issues for OPAT.
Then, we decided to focus our manuscript on patients' related outcomes as frailty, body weight and cognitive status, and to deeper analyze the microbiological and infectious aspects in an other study, that is ongoing.
As the aspects you pointed is very interesting, we preferred not to dismiss the topic briefly.